# Influence of Fines Content and Pile Surface Characteristics on the Pullout Resistance Performance of Piles

**DOI:** 10.3390/ma17010124

**Published:** 2023-12-26

**Authors:** Seungkyong You, Kwangwu Lee, Gigwon Hong

**Affiliations:** 1Department of Civil Engineering, Myongji College, 134 Gajwa-ro, Seodaemun-gu, Seoul 03656, Republic of Korea; yousk@mjc.ac.kr; 2Department of Geotechnical Engineering Research, Korea Institute of Civil Engineering and Building Technology, Goyang-si 10223, Republic of Korea; 3Department of Civil Engineering, Halla University, 28 Halladae-gil, Wonju-si 26404, Republic of Korea

**Keywords:** soil–pile interface, shear resistance, interface friction angle, adhesion, pullout resistance

## Abstract

In this study, the direct shear test and model pullout test results are presented to assess the impact of soil fines content and shear resistance characteristics of the pile–soil interface on the pullout resistance of drilled shafts. The direct shear test on the soil–pile interface was conducted based on the pile surface simulated using sandpaper with three roughness types (#24, #40, and #400) and varying fines content. The direct shear test results of soil showed that the internal friction angle decreased by about 29% and the cohesion increased by about 110% when the fine powder content increased from 5% to 30%. Specifically, in the case of soil–sandpaper (#24), the interface friction angle decreased by about 31%, and the adhesion increased by about 16%. The sandpaper with a roughness of #40 and #400 also showed a similar trend. Normalizing the shear strength parameters from the direct shear test demonstrated an intersection between the normalized curves of the friction angle and cohesion (or adhesion) within a specific fines content range. This suggests that shear strength parameters play a significant role based on fines content. Analyzing the normalized index using model pullout test results indicated the necessity to evaluate the contribution of friction angle and cohesion (or adhesion) of the shear surface, taking into account the fines content of the soil for predicting pile pullout resistance.

## 1. Introduction

Pile foundation is a fundamental structure designed to support various loads in situations where the upper structure’s load cannot be sustained by soft ground. This construction method is employed to transmit loads to a deep supporting layer when direct foundation installation is challenging due to a high groundwater level or when a high concentrated load is present.

Pile foundation is categorized into steel, PHC, and drilled shafts based on the material used, and it is engineered to withstand the compressive load from the upper structure. However, for pile foundations subjected to pullout loads, the friction resistance at the pile–soil interface becomes crucial for structural stability. Evaluating the shear resistance characteristics of the pile–soil interface is particularly vital for the pullout resistance of drilled shafts, predominantly composed of concrete. Given that dynamic loads resulting from earthquakes can accelerate the pullout of pile foundations along with ground deformation, assessing the pullout resistance is a crucial evaluation factor.

In the realm of seminal investigations pertaining to the pullout behavior of pile foundations, refs. [1,2] have proposed calculation formulas for pullout forces. These formulas are grounded in the analysis that attributes the pullout resistance of piles to the skin friction manifesting at the pile–soil interface. Building upon the Meyerhof theory, ref. [3] introduced an experimental formula derived from model tests, incorporating the concept of the limit depth. Furthermore, ref. [4] contributed significant research findings encompassing the stability and design of piles, with a specific focus on the pullout resistance characteristics of drilled shafts. Ref. [5] proposed a formula to predict the ultimate bearing capacity of uplift piles in combined soil and rock mass.

In recent years, various experimental and numerical analysis studies have been conducted on the behavior of pile foundation and the evaluation of material–soil interface characteristics considering climate change and various structure construction conditions. Ref. [6] evaluated the friction resistance of the sleeve installed at the bottom of piles through laboratory experiments to improve the bearing capacity of the pile foundation installed in sandy soil. Ref. [7] analyzed the change in the vertical load transmitted to a single pile and the validity of the bearing capacity calculation method through 3D numerical analysis and experiments when the pile is subjected to vibratory loads. Ref. [8] conducted load tests on the medium-sized steel pile foundation and concrete pile foundation constructed in marine clay and evaluated the time change–bearing capacity relationship based on the results. Refs. [9,10] analyzed the effect of grouting on the improvement in the shear strength of the pile–soil interface and the soil around the pile foundation. Ref. [11] conducted research on the influence of grouting in the soil around the pile foundation on the end bearing capacity of the pile foundation.

Ref. [12] estimated that the liquefaction of the soil around piles has a significant impact on dynamic interactions between the soil and piles and proposed an interface model considering the pile–soil interface friction angle. They verified the validity of the proposed model through a comparison with centrifuge test results. Ref. [13] conducted an experimental study to evaluate the axial resistance effect of piles that resist pullout. Ref. [14] reported that the pullout resistance of the piles constructed in soil significantly decreases as the groundwater level increases. Ref. [15] reported that the resistance of pile foundation under the pullout load is affected by the relative density of sand. Ref. [16] proposed a load transfer mechanism model predicated on friction resistance at the pile–soil interface for pullout-resistant piles in multi-layered soil. Ref. [17] emphasized the necessity of incorporating friction resistance into evaluations of ultimate pullout resistance based on model test outcomes. Ref. [18] conducted tensile load tests on model piles with varying surface roughness in sandy soil, stressing the importance of considering surface roughness for calculating pile bearing resistance. Ref. [19] investigated the correlation of pile–soil interface for screw piles, noting that soil shear strength and geometric variables of the screw must be considered for the ultimate bearing capacity. Ref. [20] presented a method for calculating the ultimate skin resistance of screw piles, validated through model and field tests. Ref. [21] evaluated screw pile performance under axial loads through laboratory tests, digital image correlation (DIC), and discrete element modeling (DEM), examining the influence of soil conditions on pile performance. Ref. [22] analyzed the load transfer characteristics of the pile–rock interface, establishing a model for calculating pile bearing capacity based on friction resistance. Ref. [23] developed and verified a prediction model for the socket shaft resistance of piles in rock layers in the Dubai area.

Further contributing to this body of knowledge, Ref. [24] employed finite element analysis to assess the pullout performance of helical piles, proposing a formula for helical pile pullout resistance. Ref. [25] evaluated factors influencing pullout resistance in clay soil where undrained cohesion increases linearly with depth. Ref. [26] assessed the reliability of algorithms predicting pile pullout behavior, while ref. [27] evaluated adhesion through friction resistance at the pile–soil interface according to various design standards, drawing insights from field cases.

Ref. [28] systematically assessed friction resistance at the material–soil interface, elucidating friction coefficients through shear tests involving diverse materials and soil types. Ref. [29] delved into friction resistance evaluations via sand and concrete friction experiments conducted under cyclic loads. In a significant large-scale endeavor, ref. [30] conducted direct shear tests, proposing shear resistance and shear coefficients for the concrete–soil interface. Their contribution extended to the formulation of a rigid plastic model to elucidate interface deformation. Ref. [31] specifically measured the friction angle at the soil–pile interface, correlating it with estimations of pile bearing capacity. Addressing the broader context of the soil–structure interaction, ref. [32] underscored the pivotal role of the soil–structure interface model. They introduced a model incorporating various parameters to discern the mechanical characteristics of the soil–structure interface. The model’s validity was established through meticulous comparisons with direct shear test results. In a related exploration, ref. [33] conducted shear tests, suggesting a correlation between the occurrence rate of peak interface friction and the particle size of sand in the interface between sand with varying roughness values and steel plates. Acknowledging the multifaceted nature of the soil–structure interface, ref. [34] emphasized the necessity for comprehensive studies, considering varying soil conditions, test characteristics (e.g., test devices), and analytical parameters (e.g., applied models). Ref. [35] reported that the resistance characteristics of the shear surface formed along the pile–soil interface are contingent upon the pile surface roughness, initial soil density, and resistance stress of the pile. Ref. [36] contributed direct shear test results to illuminate the shear characteristics of the pile–soil interface, recognizing the pivotal influence of concrete–soil interface characteristics on pile skin friction resistance. Meanwhile, ref. [37] investigated failure loads for the soft soil–concrete interface, assessing friction behavior and stiffness alterations. Addressing sand–concrete interface mechanics, ref. [38] systematically evaluated the effects of relative density and roughness of sand on the mechanical characteristics of the sand–concrete interface. Studies have been conducted on the strength characteristics of mixed soil. Ref. [39] reported that at the same granular void ratio of mixed soil, plastic fines generally decrease the undrained strength, and non-plastic fines increase the undrained strength. Ref. [40] investigated the possibility of estimating shear strength using an established method for binary mixtures to overcome the difficulties of undisturbed sampling.

Drilled shafts are recognized for their effectiveness in pullout resistance compared to other pile types. When installed in soil, they can achieve economic efficiency by ensuring adequate friction resistance from the pile skin. Consequently, numerous drilled shafts with varying diameters have been employed to attain both end bearing capacity and friction bearing capacity based on the soil characteristics at construction sites. However, previous studies have been insufficient in evaluating the shear resistance at the pile–soil interface and establishing the correlation between shear resistance and pullout resistance based on soil components.

This study aims to address this gap by comprehensively evaluating the impact of fines content in soil and the shear resistance characteristics of the pile–soil interface on the pullout resistance of drilled shafts. The assessment is conducted through a rigorous examination utilizing both the direct shear test and model pullout test results, as elucidated by [41]. 

## 2. Materials and Methods

### 2.1. Pullout Resistance of Pile

The assessment of pile pullout resistance primarily relies on pullout load tests, compressive loading tests, and empirical formulas. Among these, pullout load test results are generally considered the most reliable. In cases where pullout resistance cannot be directly obtained through load tests, an alternative approach involves evaluating pullout resistance based on the skin friction of the pile, often determined through compressive loading tests. Both of these methodologies necessitate on-site field tests. However, in situations where field tests pose logistical challenges, the evaluation of pullout resistance resorts to laboratory model tests. The pullout coefficient, derived from a pullout resistance calculation formula, is then applied in the design process. Essentially, the ultimate pullout resistance of a pile is expressed as the sum of pure pullout resistance and the weight of the pile. Ref. [1] posited that the pure pullout resistance of a pile in soil hinges on the friction force between the soil and the pile surface. Consequently, the calculation formula for pile pullout resistance, as represented by Equation (1), underscores the critical importance of shear resistance, encompassing adhesion and the friction angle, at the pile–soil interface in deriving the pullout coefficient from this equation:(1)$$P_{u}=\left(C_{a}+{p’}_{o}^{\;}\;K_{u}\;tan\delta \right)\;A_{s},$$
where $P_{u}$ is the pullout resistance, $C_{a}$ is the pile–soil interface adhesion, ${p’}_{o}^{\;}$ is the effective vertical stress, $K_{u}$ is the pullout coefficient, δ is the pile–soil interface friction angle, and $A_{s}$ is the area of the pile surface.

### 2.2. Laboratory Experiment

In this study, the direct shear test, aimed at assessing the shear strength at the pile–soil interface, was conducted to evaluate the impact of fines content in soil and the shear resistance at the pile–soil interface on the pullout resistance of the pile. Additionally, the correlation between the shear resistance characteristics determined through the direct shear test and the findings of the model pullout test conducted earlier was analyzed [41].

#### 2.2.1. Engineering Properties of Soils

The soils utilized in both the direct shear test and the model pullout test comprised Jumunjin standard sand and fine-grained soil. The grain size distribution curves of Jumunjin standard sand and fine-grained soil are illustrated in Figure 1, while Table 1 provides a summary of soil properties categorized by soil type. In accordance with the Unified Soil Classification System (USCS), Jumunjin standard sand and fine-grained soil were classified as poorly graded sand (SP) and low-compression silt (ML), respectively. 

In this study, a series of lab-scale tests were conducted to assess the impact of fines content in soil on the pullout resistance of piles. Specifically, Atterberg limit tests were performed to evaluate the engineering properties of soil based on the fines content. The Atterberg limit test for a mixture of sand and fine-grained soil was performed with soil samples that passed through a No. 40 sieve (standard sieve size 0.425 mm), which contained a certain percentage of fine sand. Figure 2 and Table 2 present the Atterberg limit test results and the soil classification outcomes according to the plasticity chart. The fines content means the percentage of fine-grained soil that passes the 200 sieve of about 85% or more contained in the sand. In addition, the fines content was applied as the weight ratio of the fine-grained soil contained in the sand.

The test outcomes reveal that when the fines content was 5%, the soil was classified as SP, aligning with the Jumunjin standard sand. As the fines content increased to a range of 10 to 18%, the soil was categorized as SC-SM, and for a fines content of 19% or higher, it was classified as SC.

#### 2.2.2. Direct Shear Tests 

This study involved a direct shear test on soil, with variations in fines content. Additionally, a direct shear test was performed on the soil–pile interface, considering the simulated pile surface created using sandpaper with three distinct roughness levels (#24, #40, and #400), along with variations in fines content. The direct shear test of mixed soil (DS-OO) was performed in 10 cases, and the shear resistance evaluation of the soil–pile interface (DSSP-OO) was performed in a total of 7 cases with the fines content at 5% intervals. The specifics of the test variations are presented in Table 3.

As shown in Figure 3, the direct shear test used a large-scale direct shear test device equipped with a large-scale shear box (size 0.3 m (B) × 0.3 m (L)) to sufficiently demonstrate the shear resistance of the soil–pile interface. The direct shear test device consists of separate upper and lower shear boxes, a vertical load loading device, and a measurement device to check stress and deformation. In order to simulate the pile surface in the direct shear test, a block model with sandpaper attached was placed in the lower shear box. Three types of sandpaper (#24, #40, #400) were used to evaluate the shear characteristics according to the surface roughness of the pile. ‘#’ is a symbol representing the grit of the sandpaper surface, and the ‘number’ is the number of particles attached to a unit area (1in × 1in). The diameters of grit 24, 40, and 400 are 0.764 mm, 0.425 mm, and 0.035 mm, respectively. In other words, this means that sandpaper with small numbers is relatively rough. Figure 4a,b show the sandpaper attached to the block and the sandpaper surface of three types, respectively.

As shown in Figure 5, the direct shear test method can be summarized as follows: (i) Place a block with sandpaper attached that simulates the pile surface in the lower shear box. (ii) Mix Jumunjin standard sand and fine-grained soil considering the weight ratio. (iii) Place and compact the soil sample in 3 layers (60 mm/layer) in the upper shear box. (iv) Install a vertical loading device (with rubber membrane) on the model ground and install a measurement device to check shear stress and deformation. (v) After applying vertical loading with air pressure within the rubber membrane, shear and measure using a strain control of 1 mm/min. For reference, the mixed soil for the model ground was mixed using a stirring device for a long time to ensure that the Jumunjin standard sand and fine-grained soil were evenly distributed.

#### 2.2.3. Pullout Tests of Model Pile Wrapped in Sandpaper

Furthermore, this study undertook an analysis of the correlation between the shear resistance characteristics obtained from the direct shear test and the results derived from a previous model pullout test [41]. While the model pullout test is extensively detailed in the aforementioned prior study, a brief overview is provided herein.

In the model pullout test, the outcomes are influenced by both the dimensions of the experimental setup and the size and conditions of the model pile. To determine the size of the experimental setup that can consider the influence range of the surrounding soil and the model pile size when vertical and pullout loads are applied to the pile, a previous study was referenced [42]. A soil box with a diameter of 0.28 m and a height of 0.56 m was prepared. Its inner wall was treated with chrome to mitigate friction. The model pile, designed for the pullout test, possessed a diameter of 0.05 m and a length of 0.4 m. To simulate a drilled shaft, the model pile, crafted from steel, was uniformly encased with sandpaper, ensuring a consistent friction resistance at the model pile–soil interface. Figure 6 shows the pullout test device and model pile. 

The model soil was created considering the fines content and relative density of the soil, with the drawing speed of the model pile set at 1 mm/min. The model soil according to relative density was created by inversely calculating the required unit weight for each relative density condition using the results of the maximum and minimum dry unit weight tested for each fines content.

The pullout load and displacement were measured using load cells and LVDTs until the model pile exhibited a pullout displacement of 40 mm. Table 4 shows the various cases considered in the model pullout test.

## 3. Results and Discussion

### 3.1. Shear Resistance of Soils and Pile–Soil Interface

#### 3.1.1. Test Results

Figure 7 and Table 5 show the results obtained from the direct shear test. First, the soil test results revealed a decrease in the internal friction angle and an increase in cohesion with a rise in fines content. As the fines content was elevated from 5% to 30%, the internal friction angle of the soil exhibited a reduction of about 29%, while apparent cohesion experienced an increase of approximately 110%.

The soil–sandpaper direct shear test results were categorized based on sandpaper roughness types. With a roughness of #24 (high roughness), there was a tendency for the interface friction angle to decrease, accompanied by an increase in adhesion due to the elevated fines content. Specifically, the interface friction angle exhibited a reduction of approximately 31%, while the adhesion increased by approximately 16%. This pattern persisted with sandpaper roughness values of #40 and #400. At #40 (medium roughness), the interface friction angle decreased by about 34%, and the adhesion increased by 15%. At #400 (low roughness), the interface friction angle experienced a reduction of approximately 52%, coupled with an 8% increase in adhesion.

In essence, an increase in fines content within the soil led to an escalated reduction rate in the interface friction angle and a diminished rate of increase in adhesion. 

#### 3.1.2. Shear Resistance Characteristics of Soil–Pile Surface Interface with Fines Content

The direct shear test results were employed to investigate the impact of fines content on the shear resistance at the soil–pile surface interface. To assess the fines content’s influence on strength parameters under various interface conditions, the 5–30% fines content range was applied for analysis.

As depicted in Figure 8, regression curves were fitted using power equations to express the soil’s internal friction angle and the friction angle at the soil–pile surface interface. The cohesion of the soil and adhesion at the soil–pile surface interface were characterized using quadratic functional equations. These equations were employed to ensure the reliability of predictions.

Comparison of shear resistance at the soil–pile surface interface and the shear resistance characteristics of the soil under identical fines content conditions revealed intriguing patterns. Figure 8a illustrates that, for the same fines content, soil exhibits low cohesion but a high internal friction angle. Conversely, regardless of sandpaper roughness conditions, adhesion was consistently high, even when the shear resistance at the soil–pile surface interface displayed a relatively low interface friction angle. For instance, when the fines content stood at 5%, the interface friction angle constituted approximately 75 to 90% of the internal friction angle. However, adhesion values were notably elevated, ranging from 810 to 980% of cohesion. This observation underscores the substantial dependence of soil shear resistance on the internal friction angle. Importantly, both the interface friction angle and adhesion must be considered when assessing shear resistance at the soil–pile surface interface.

In their study, Ref. [41] provided a quantitative identification of the internal friction angle and cohesion characteristics of soil based on fines content, presenting Equations (2) and (3) that enable normalization using regression equations. In the present study, this methodology was adopted to formulate normalized index equations for the shear strength parameter at the soil–pile surface interface, represented by Equations (4) and (5).
(2)$$N_{i(\emptyset )}=1-\left(\frac{\emptyset_{\left(max\right)}-\emptyset_{\left(f.c.\right)}}{\emptyset_{\left(max\right)}-\emptyset_{\left(min\right)}}\right),\;\;(0\;\leq \;N_{i\left(\emptyset \right)}\;\leq 1)$$
(3)$$N_{i(c)}=1-\left(\frac{c_{\left(max\right)}-c_{\left(f.c.\right)}}{c_{\left(max\right)}-c_{\left(min\right)}}\right),\;\;(0\;\leq \;N_{i\left(c\right)}\;\leq 1)$$
(4)$$N_{i(\delta )}=1-\left(\frac{\delta_{\left(max\right)}-\delta_{\left(f.c.\right)}}{\delta_{\left(max\right)}-\delta_{\left(min\right)}}\right),\;\;(0\;\leq \;N_{i\left(\delta \right)}\;\leq 1)$$
(5)$$N_{i(c_{a})}=1-\left(\frac{c_{a(max)}-c_{a\left(f.c.\right)}}{c_{a(max)}-c_{a(min)}}\right),\;\;(0\;\leq \;N_{i(c_{a})}\;\leq 1)$$

The shear strength parameters corresponding to fines content are depicted in normalized curves, utilizing Equations (2)–(5), as illustrated in Figure 9. Notably, the intersection between the normalized curves of the friction angle and cohesion (or adhesion) was identified within the fines content range of 14–17%, irrespective of the shear surface characteristics based on the material. This observation signifies that shear resistance is influenced by the friction angle and cohesion (or adhesion), contingent upon the fines content.

### 3.2. Influence of Fines Content on the Pullout Resistance Performance of Pile

The impact of fines content on pile pullout resistance was assessed using the outcomes derived from the model pullout test conducted by [41]. In the model pullout test, fines contents ranging from 5% to 20% were incorporated into the model soil, consistent with the soil utilized in this study. For fines content exceeding 19%, the soil classification resulted in SC, enabling the evaluation of shear resistance characteristics based on fines content.

The key findings from the model pullout test by [41] can be succinctly summarized. Irrespective of the relative density and fines content, the maximum pullout resistance was attained when the pile’s pullout displacement ranged between 3 and 6 mm. Following the achievement of the maximum pullout resistance, the residual strength was sustained even as the pullout resistance diminished. Notably, the magnitude of the residual strength remained consistent, irrespective of variations in relative density and fines content. Figure 10 provides a visual representation of the model pullout test results. 

The analysis revealed that the pullout resistance of the pile exhibited an increase with higher relative density under the same fines content conditions. Across all relative density conditions, the regression curve of pullout resistance displayed a tendency to decrease within the fines content range of 5 to 13%. However, beyond a fines content of 13%, the pullout resistance increased, with the rate of increase amplifying alongside higher relative density.

These findings find support in the direct shear test results for each fines content. The pivotal contributors to the development of pile pullout resistance are the shear strength parameters of the surrounding soil (internal friction angle and cohesion) and those of the soil–pile surface interface (interface friction angle and adhesion). As elucidated in Figure 9b–d, these contributing factors exhibit opposing effects within the fines content range of approximately 14 to 17%. This suggests that the pullout resistance characteristics of piles align more closely with the shear resistance characteristics of the soil–pile surface interface than with the shear resistance characteristics of the soil. In other words, when the fines content is below the 14–17% range, the pullout resistance of piles decreases in tandem with the diminishing interface friction angle. This dependency arises due to the magnitude of the interface friction angle. Conversely, when the fines content surpasses the 14–17% range, the pullout resistance of piles increases, correlating with the rising influence of adhesion, which significantly impacts pile pullout resistance.

To reiterate, the evaluation of influential factors such as the friction angle and cohesion (or adhesion) from the normalized index based on fines content, as utilized in Figure 9, demonstrates that the fines content of the soil contributes significantly to the pullout resistance of piles. 

## 4. Conclusions

This study aimed to assess the impact of soil fines content and the shear resistance characteristics of the pile–soil interface on the pullout resistance of drilled shafts, utilizing data from the direct shear test and model pullout test. The conclusions drawn from this investigation can be succinctly summarized as follows:

(1)The direct shear test outcomes revealed that, irrespective of the material characteristics of the shear surface, the internal friction angle of soil and the soil–sandpaper interface friction angle decreased with increasing fines content. Conversely, the cohesion of soil and the adhesion at the soil–sandpaper interface exhibited an upward trend as the fines content increased. Remarkably, the increase in the rate of adhesion at the soil–sandpaper interface was approximately ten times higher than that of cohesion. It was further emphasized that both the interface friction angle and adhesion are crucial considerations in assessing the shear resistance characteristics at the soil–sandpaper interface, as compared to the shear resistance of soil.(2)The shear strength parameters, normalized based on the fines content using the direct shear test results, revealed an intersection between the normalized curves of the friction angle and cohesion (or adhesion) within a specific fines content range, irrespective of the shear surface characteristics of the material. This implies that shear strength parameters serve as influential factors dependent on fines content.(3)Analysis of the normalized index results from the model pullout test indicated a substantial contribution of the soil fines content to the pullout resistance of the pile. Consequently, predicting the pullout resistance of piles necessitates an evaluation of the contribution of the friction angle and cohesion (or adhesion) in consideration of the fines content of the soil, aligning with the methodology employed in this study.(4)Piles are mostly installed in saturated soil at the bottom of the ground surface. This research could not consider the conditions of saturated soil because it is difficult to simulate the conditions of model ground with saturated soil using the direct shear test device applied in this research. In other words, this research includes test results due to the limitations of the testing device. Therefore, the research considering the conditions of saturated soil must continue to be conducted in order to ensure the reliability of the shear resistance evaluation results of the soil–pile interface.

## Figures and Tables

**Figure 1 materials-17-00124-f001:**
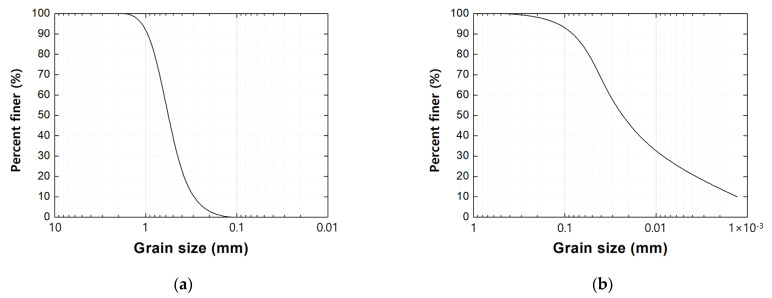
Grain size distribution curve of soils: (**a**) Jumunjin standard sand; (**b**) fine-grained soil.

**Figure 2 materials-17-00124-f002:**
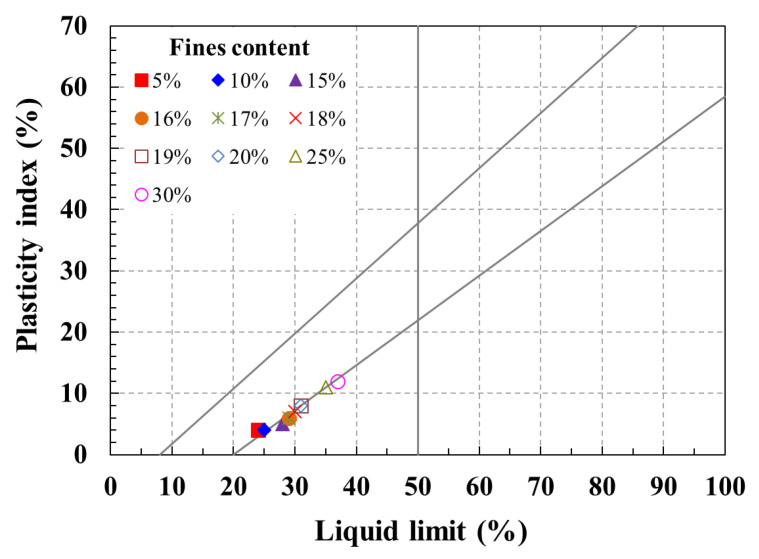
Atterberg limit test results according to fines content of soils.

**Figure 3 materials-17-00124-f003:**
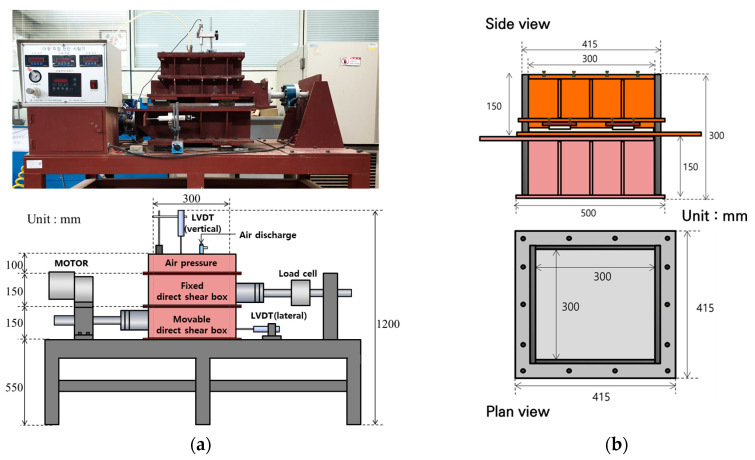
Large-scale direct shear test device: (**a**) photo and schematic diagram of device; (**b**) schematic diagram of direct shear box.

**Figure 4 materials-17-00124-f004:**
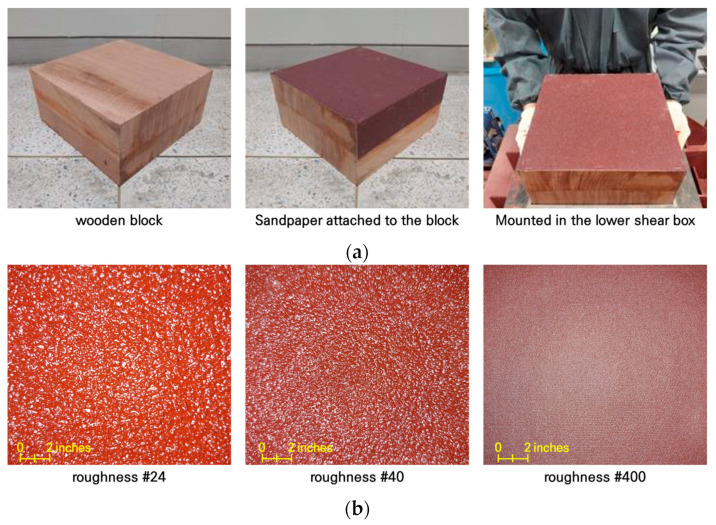
Surface description of piles: (**a**) shear block for surface description of piles; (**b**) sandpaper type.

**Figure 5 materials-17-00124-f005:**
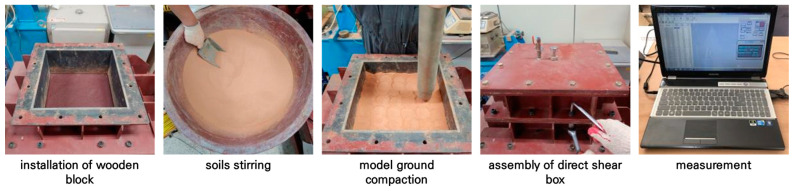
Direct shear test procedure.

**Figure 6 materials-17-00124-f006:**
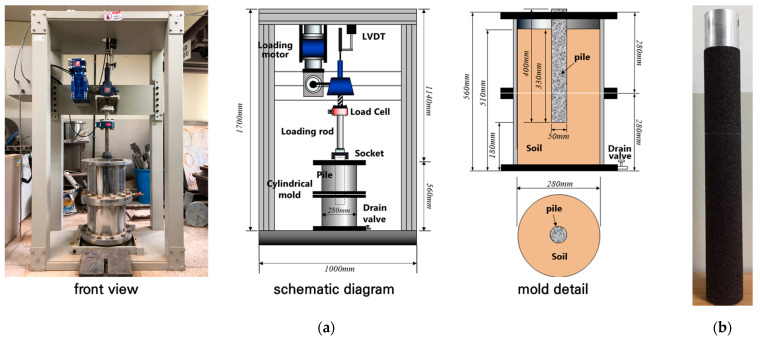
Overview of pullout test [41]: (**a**) photo and schematic diagram of device; (**b**) model pile with sandpaper.

**Figure 7 materials-17-00124-f007:**
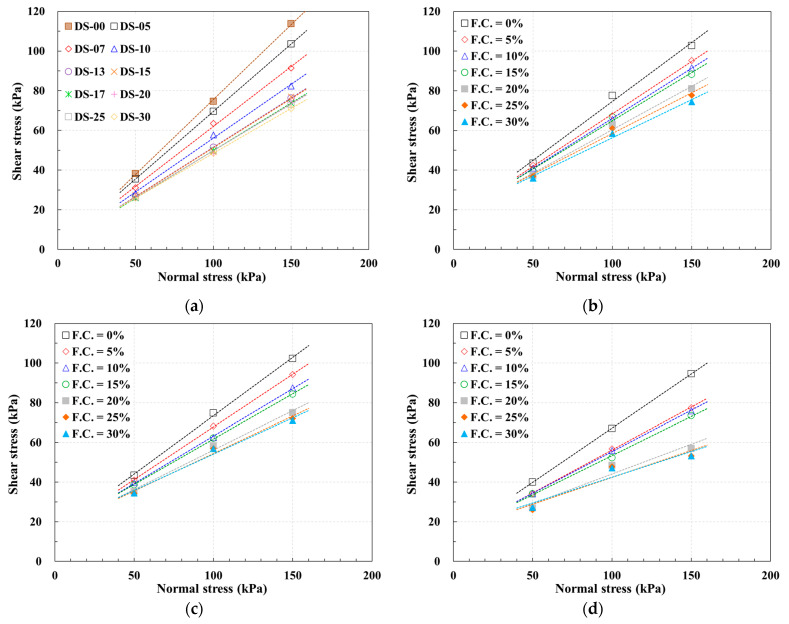
Direct shear test results: (**a**) soils; (**b**) soil–sandpaper (#24); (**c**) soil–sandpaper (#40); (**d**) soil–sandpaper (#400).

**Figure 8 materials-17-00124-f008:**
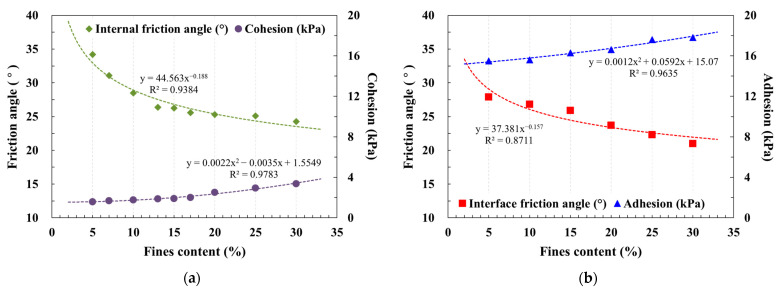
Relationship of fines content and shear strength parameters: (**a**) soils; (**b**) soil–sandpaper (#24); (**c**) soil–sandpaper (#40); (**d**) soil–sandpaper (#400).

**Figure 9 materials-17-00124-f009:**
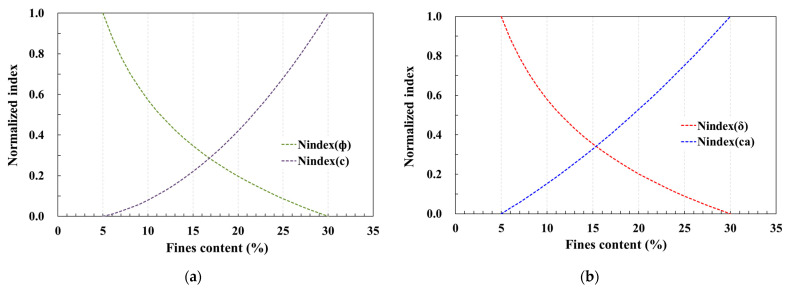
Relationship of fines content and normalized index based on shear strength parameters: (**a**) soils; (**b**) soil–sandpaper (#24); (**c**) soil–sandpaper (#40); (**d**) soil–sandpaper (#400).

**Figure 10 materials-17-00124-f010:**
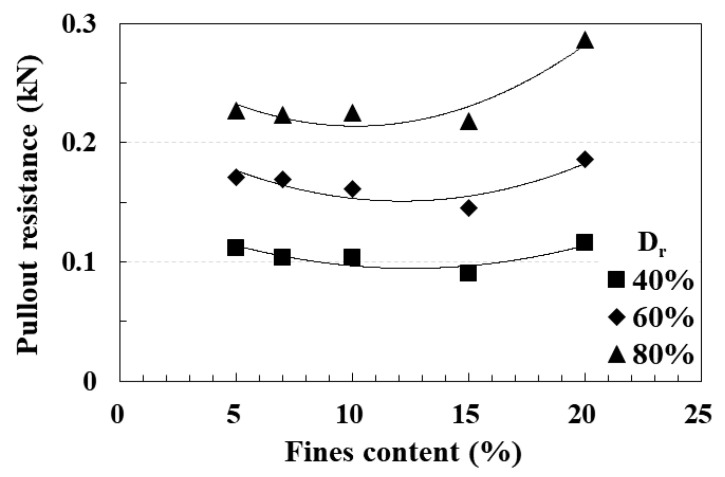
Relationship between fines content and pullout resistance in terms of relative density [41].

**Table 1 materials-17-00124-t001:** Soil properties.

Soil Classification	Properties	
Jumunjin standard sand	Gs	2.63
	Cu	2.1
	Cg	1.1
	U.S.C.S.	SP
Fine-grained soil	LL (%)	39.4
	PL (%)	31.3
	PI (%)	8.1
	U.S.C.S.	ML

**Table 2 materials-17-00124-t002:** Soil classification according to the Atterberg limit.

Fines Content (%)	PI (%)	LL (%)	PL (%)	Soil Classification
5	4	24	20	SP
10	4	25	21	SP-SM
15	5	28	23	SC-SM
16	6	29	23	SC-SM
17	6	29	23	SC-SM
18	7	30	23	SC-SM
19	8	31	23	SC
20	8	31	23	SC
25	11	35	24	SC
30	12	37	25	SC

**Table 3 materials-17-00124-t003:** Direct shear test cases.

Test Classification			Test Condition	
Soils	Soil–Sandpaper		Fines Content (%)	Normal Stress (kPa)
DS-00	Sandpaper type #24, #40, #400	DSSP-00	0	50 100 150
DS-05		DSSP-05	5	
DS-07			7	
DS-10		DSSP-10	10	
DS-13			13	
DS-15		DSSP-15	15	
DS-17			17	
DS-20		DSSP-20	20	
DS-25		DSSP-25	25	
DS-30		DSSP-30	30	

Note (test classification): DS-OO: direct shear test—fines content. DSSP-OO: direct shear test with sandpaper—fines content.

**Table 4 materials-17-00124-t004:** Pullout test cases of model pile [41].

Cases	Fines Content (%)	Relative Density (%)	Buried Depth of Pile (m)
P-05-40	5	40	0.33
P-05-60		60	
P-05-80		80	
P-07-40	7	40	
P-07-60		60	
P-07-80		80	
P-10-40	10	40	
P-10-60		60	
P-10-80		80	
P-15-40	15	40	
P-15-60		60	
P-15-80		80	
P-20-40	20	40	
P-20-60		60	
P-20-80		80	

**Table 5 materials-17-00124-t005:** Shear strength parameters.

Fines Content (%)	Cases DS (Soils)		Case DSSP (Soil–Sandpaper)					
	Internal Friction Angle (ϕ, °)	Cohesion (c, kPa)	Interface Friction Angle (δ, °)			Adhesion (C_a_, kPa)		
			#24	#40	#400	#24	#40	#400
0	37.1	0.0	30.7	30.5	28.6	15.3	14.7	12.6
5	34.2	1.6	27.9	27.9	23.4	15.5	14.8	12.8
7	31.1	1.7						
10	28.5	1.8	26.8	25.5	22.7	15.6	15.5	13.4
13	26.4	1.9						
15	26.3	1.9	25.9	24.5	21.6	16.3	16.1	13.8
17	25.6	2.0						
20	25.3	2.5	23.7	21.7	16.7	16.6	16.4	14.1
25	25.1	3.0	22.3	20.7	15.2	17.6	16.6	15.2
30	24.3	3.4	21.0	20.1	14.6	17.8	17.6	16.5

## Data Availability

Data are contained within the article.

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
