# Peer review of "Influence of Fines Content and Pile Surface Characteristics on the Pullout Resistance Performance of Piles"

_materials, 2023, doi:10.3390/ma17010124_

Round 1

Reviewer 1 Report

Comments and Suggestions for Authors

The study entitled Influence of fines content and pile surface characteristics on the pullout resistance performance of piles presents, in its current version, some problems that impair its publication potential.

It is indicated that experimental tests were carried out, however, these are described very briefly. Please describe your experimental test in more detail and show images and schematic diagrams of it to improve understanding of your article.

Kind regards

Reviewer 2 Report

Comments and Suggestions for Authors

The author evaluated the influence of the shear resistance characteristics of the pile soil interface on the drilling pull-out resistance through direct shear tests and model pull-out tests, but there are still many issues that need to be revised in the research.

1. The content in the abstract is too general, and the key content should be written more specifically.

2. In the introduction, some relevant literature should be cited, such as:

A method for calculating the ultimate bearing capacity of lift piles in combined soil and rock mass. European Journal of Environmental and Civil Engineering, 26 (6), 2158-2183

3. The meanings represented by symbols such as DS-05 # 40 and DSSP-10 in Table 3 should be explained in advance in the text.

4. In section 2.2.1, the author only explained the use of two types of soil, but did not explain how to mix the two types of soil evenly during the experiment. I have a question, when sand and soil are mixed, it is difficult for fine particles to be evenly distributed inside. How does the author ensure that sand and soil are mixed together?

5. The author mentioned that the research in this article is based on reference [39]. Since I am unable to download reference [39], do all the experimental data used in this article appear in reference [39]?

6. In section 2.2.2. Direct shear tests, the authors mentioned using sandpaper with three different roughness levels to simulate pile surfaces. What are the roughness levels of the three sandpaper levels? Please provide an explanation

7. the Direct shear tests and Pullout tests in the text should provide a flowchart to illustrate the process

8. Section 3.2, the author should provide pullout resistance for piles with different roughness levels.

9. The author should provide the scaling basis for the model experiment in this article

Comments on the Quality of English Language

The author evaluated the influence of the shear resistance characteristics of the pile soil interface on the drilling pull-out resistance through direct shear tests and model pull-out tests, but there are still many issues that need to be revised in the research.

1. The content in the abstract is too general, and the key content should be written more specifically.

2. In the introduction, some relevant literature should be cited, such as:

A method for calculating the ultimate bearing capacity of lift piles in combined soil and rock mass. European Journal of Environmental and Civil Engineering, 26 (6), 2158-2183

3. The meanings represented by symbols such as DS-05 # 40 and DSSP-10 in Table 3 should be explained in advance in the text.

4. In section 2.2.1, the author only explained the use of two types of soil, but did not explain how to mix the two types of soil evenly during the experiment. I have a question, when sand and soil are mixed, it is difficult for fine particles to be evenly distributed inside. How does the author ensure that sand and soil are mixed together?

5. The author mentioned that the research in this article is based on reference [39]. Since I am unable to download reference [39], do all the experimental data used in this article appear in reference [39]?

6. In section 2.2.2. Direct shear tests, the authors mentioned using sandpaper with three different roughness levels to simulate pile surfaces. What are the roughness levels of the three sandpaper levels? Please provide an explanation

7. the Direct shear tests and Pullout tests in the text should provide a flowchart to illustrate the process

8. Section 3.2, the author should provide pullout resistance for piles with different roughness levels.

9. The author should provide the scaling basis for the model experiment in this article

Reviewer 3 Report

Comments and Suggestions for Authors

The paper aims to investigate the influence of the fines content on the pull-out resistance of piles with non-smooth surfaces.

The article is interesting and well-written.

There are a few details that are missing for being worthy of publication:

1) How did you define the fines? What grain radius is considered? Please add it to the text

2) Starting from the particle size distribution functions of Figure 1, what approach did you use to increase the fines content? It is not clear if the PSD functions are shifted maintaining the same shape or if just the tail corresponding to the finer has been modified. It could be worth enriching Figure 1a-b by adding the corresponding investigated curves.

3) A "direct shear test was performed on the soil–pile interface" is not clear. In a direct shear test only the soil is investigated. The methodology lacks of clarity about the types of tests conducted. Photos of the experiments are missing and/or schematics of what has been conducted

4) It is not clear if the roughness of the sandpaper is really representing a realistic scenario. Considering the scaling factors involved, what is the actual representation of the 3 grades used for the experiment?

5) Regarding the pullout test, how did the Authors differentiate the effect of the cohesion from the internal frictional angle? And how the pullout coefficient Ku has been evaluated?

Comments on the Quality of English Language

Quality is acceptable and I do not have concerns about

Reviewer 4 Report

Comments and Suggestions for Authors

The paper presents a number of direct shear tests on reconstituted samples with different proportion of fines and some pullout tests of model piles aimed at evaluating the relevance of the fines content in the pullout resistance of drilled piles. The study is particularly focused on the behaviour of the soil-pile interface.

The study is interesting and includes a large number of tests and data. However, some aspects of the experiments are unclear to me and, as a consequence, I have some concerns about the soundness of the conclusions.

First of all, I note that the investigation of mechanical behaviour varying fines content is the approach followed in the literature for the study of mixtures and the entire test scheme is very similar to a classical study for binary mixtures. Surprisingly, any reference to this field of research is considered in the paper. I suggest the following papers just to introduce the topic, then the Authors themselves will be able to find the more interesting papers by checking the references:

-        Ni, Q., Tan, T. S., Dasari G. R., Hight, D. W. (2004). Contribution of fines to the compressive strength of mixed soils. Géotechnique 54(9): 561-569.  

-        Ruggeri, P., Segato, D., Fruzzetti, V. M. E., & Scarpelli, G. (2016). Evaluating the shear strength of a natural heterogeneous soil using reconstituted mixtures. Géotechnique, 66(11), 941-946.

Secondly, I have some concern about the choice of the granular composition of the fine-grained soil in the presented study. While the Joomunjin is a monogranular sand very common for this kind of tests, I am surprise to see that the fine-grained soil is a sandy silt with clay (roughly 15% clay, 65% silt and 30%sand), characterized by low plasticity. The behaviour of silt is difficult to understand because it is not clear if the granular or cohesive behaviour prevail. For this reason, the studies on binary mixtures typically start by testing coarse grained mixtures (e.g. gravel-sand, coarse sand-fine sand) or cohesive mixtures (e.g. sand-clay). Of course, this choice make hard to have a clear picture of the results.

Beyond these two major aspects, following the research scheme you have presented, I have some questions:

1-     It is not clear if you performed dry or satured direct shear tests on the mixtures. My first idea is that you performed dry tests. Why that, having a cohesive part of soil in your mixtures?

2-     The amount of fine is always less than 30% and, even if it is not evaluated, I think the fines is not enough to fill the voids of the coarser soil. So, you have always tested grain-sustained mixtures, a condition in which the arrangement of fines and larger particle can modify the mechanical behaviour. How did you prepare the samples? Please explain.

3-     There is something strange in the evaluation of the Plasticity Index of the mixture (Figure 2 and Table 2). How is it possible that the mixture with 30% of fines content has the same liquid limit of the fines content alone? (i.e. LL=37 versus 39.4)? I have an hypothesis: you are measuring the Atterberg limits on the passing at sieve ASTM n.40 (d=0,425mm); so, from a side you are creating a mixture, from the other side you are removing the large part of the sand. If I am right, the Atterberg limit you evaluated cannot be associated to the mixtures. A good tentative could be scaling the results on the proportion of the passing to ASTM 40, assuming a linear relationship that is also confirmed by your data (maybe the book Head - Manual of Soil laboratory testing, volume 1, can be useful);

4-     A scheme of the direct shear test in which the soil-pile interface is tested should be presented.

5-     It is not explained why you tested different sandpapers to simulate the soil-pile interface. Also, why have not you tested a pile surface with glued a layer of the same sand you have chosen? Anyway, I think the mean diameter of the sandpaper you have used should be added. According to same research I carried out online, it seems that grit 24 = 0,764mm, 40=0,425mm, 400=0,035mm. It can help same comparison with the mean diameter of the mixtures you have tested.

6-     Table 5. It is necessary to add the shear strength resistance of pure sand and pure fine-grained material. To be honest, it is very hard to me to accept a friction angle of 24° for the mixture with 30% of fines. It is a friction angle expected for a fat clay, not for the soil you have tested. I suspect that it is a problem related to the dry execution of the direct shear test, that is not acceptable.

So, after a careful analysis, I suggest an in-depth revision of the paper and a resubmission of the manuscript.

Comments on the Quality of English Language

The english language is generally fine. Some minor editing are required.

Round 2

Reviewer 2 Report

Comments and Suggestions for Authors

Accept

Author Response

Thank you very much for the reviewer’s interest.

Reviewer 4 Report

Comments and Suggestions for Authors

Dear Authors, I have carefully read your replies to my comments and I have to confirm my doubts about the practical meaning of the shear strength measured on sand-silt mixtures in dry conditions.

Anyway, we can consider the results of the large number of experimental tests you have performed worth to be published if you clearly state the test conditions both in the “Materials and Methods” section and in the Conclusions. So, I expect to read that direct shear tests have been always performed in dry conditions (in this sense I appreciated the revised abstract in which you mentioned the term “fine powder”). Please, in the following of your research, take into account that the real piles are mostly located on saturated soils, due to the presence of a groundwater level usually not far from the ground surface.

In the following I have some suggestions to improve the quality of the paper:

1-      Atterberg limits: I understand we have to believe to test results, but I still have some concerns about the Atterberg limits of the tested soils. In your reply, you said that you have not removed the sand (roughly 15%) from the fine grained soils and that Atterberg limit tests were performed on a mixture of sand and fine-grained soil from the original sample. I understand your reply, but you have probably missed the point. It is not credible that you have been able to measure the Atterberg limits on the whole mixture and obtaining a Plastic Limit different from zero for a 95% sand mixture. So, according to the ASTM recommendation, I think you have sieved the mixture by ASTM 40 sieve and test the soil passing through it (that, of course, include a percentage of fine sand). Please, verify accurately this aspect and improve the description in the paper (i.e. L180-184 are not clear).

2-      Please, add a graph with the grading particle distribution of all the mixtures considered in Table 2.

3-      L197-198. What is the meaning of the sentence “The fines content in the soil for the direct shear test was determined based on the Atterberg limit test results”? Please, clarify

4-      It is appreciated the description of the large direct shear box you have used. Please, specify the presence (I think) of a rubber membrane at the top of the shear box, able to assure the effectiveness of the air pressure on the application of the vertical load. Large shear box should be tested to verify the reliability of the results. Can you provide some reference about the validation of the adopted shear apparatus? Can you provide some tests on the mixtures obtained by standard apparatus? (Also, I noted that this apparatus cannot work in saturated condition and this is a problem for your prospective research).

5-      Fig 4. Please add a scale bar on the pictures;

6-      Section 2.2.3 – Please, add some pictures or drawings of the pullout tests scheme;

7-      Table 4. How have you measured the relative density of the mixtures? Please, give some comments and reference to international procedure. Also, it is relevant to add the unit weight of the tested samples at the different density;

8-      Table 5. Please, limit the decimal places to one for cohesion.

9-      L259: please, consider to modify the term “cohesion” in “apparent cohesion” due to the lack of real forces between the particles you have tested;

10-   Honestly, I cannot comment section 3.1.2, it seems not very useful to me.  

Comments on the Quality of English Language

No relevant issues related to the english language are detected.
